# Comparisons of Dipstick Test, Urine Protein-to-Creatine Ratio, and Total Protein Measurement for the Diagnosis of Preeclampsia

**DOI:** 10.3390/ijerph17124195

**Published:** 2020-06-12

**Authors:** Katarzyna Stefańska, Maciej Zieliński, Dorota Zamkowska, Przemysław Adamski, Joanna Jassem-Bobowicz, Karolina Piekarska, Martyna Jankowiak, Katarzyna Leszczyńska, Renata Świątkowska-Stodulska, Krzysztof Preis, Piotr Trzonkowski, Natalia Marek-Trzonkowska

**Affiliations:** 1Department of Obstetrics, Medical University of Gdańsk, 80-214 Gdańsk, Poland; dorotaaw@gmail.com (D.Z.); padamski@gumed.edu.pl (P.A.); kasiabor1@wp.pl (K.L.); kpreis@wp.pl (K.P.); 2Department of Medical Immunology, Medical University of Gdańsk, 80-214 Gdańsk, Poland; mzielinski@gumed.edu.pl (M.Z.); karolinapiekarska18@gmail.com (K.P.); martyna830@gumed.edu.pl (M.J.); ptrzon@gumed.edu.pl (P.T.); 3Department of Neonatology, Medical University of Gdańsk, 80-214 Gdańsk, Poland; jmjassem@gmail.com; 4Department of Endocrinology and Internal Medicine, Medical University of Gdańsk, 80-214 Gdańsk, Poland; rensto@gumed.edu.pl; 5Cancer Immunology Group, International Centre for Cancer Vaccine Science, University of Gdansk, 80-214 Gdańsk, Poland; natalia.marek@gumed.edu.pl; 6Laboratory of Immunoregulation and Cellular Therapies, Department of Family Medicine, Medical University of Gdańsk, 80-214 Gdańsk, Poland

**Keywords:** preeclampsia, gestational hypertension, pregnancy, urine protein-to-creatinine ratio, proteinuria, 24 h urine sample

## Abstract

Preeclampsia affects 2–5% of pregnant women and is one of the leading causes of maternal and perinatal morbidity and mortality. We aimed to extensively evaluate proteinuria in women with preeclampsia and to determine the analytical sensitivity and specificity of and the cutoff values for urine protein-to-creatinine ratio (UPCR) and total protein in 24 h urine samples. This study included 88 women. We used the urine dipstick test, UPCR, and total protein measurement in a 24 h urine sample. The patients were divided in gestational hypertension (GH, *n* = 44) and preeclampsia (PE, *n* = 44) groups. In the GH group, 25% (11/44) of the patients presented incidentally positive results. UPCR and total protein in 24 h urine specimens were increased in the GH group compared to the PE group. Receiver operating characteristic analysis showed a UPCR cutoff of 30 mg/mmol as significant for preeclampsia, while the sensitivity and specificity were 89% (95% CI, 75–97) and 100% (95% CI, 87–100), respectively. In the 24 h urine protein test, sensitivity and specificity were 80% (95% CI, 61–92) and 100% (95% CI, 88–100), respectively, for the cutoff value of 0.26 g/24 h. In comparison to the other commonly used tests here considered, UPCR determination is a reliable, relatively faster, and equally accurate method for the quantitation of proteinuria, correlates well with 24 h urine protein estimations, and could be used as an alternative to the 24 h proteinuria test for the diagnosis of preeclampsia.

## 1. Introduction

Preeclampsia (PE) is a multisystem disorder of pregnancy and is defined as the onset of hypertension accompanied by significant proteinuria after 20 weeks of gestation [1]. PE typically affects 2–5% of pregnant women and is one of the leading causes of maternal and perinatal morbidity and mortality, especially when the condition is of early onset [1,2]. Globally, 76,000 women and 500,000 babies die each year from this disorder [1,3].

According to the International Society for the Study of Hypertension in Pregnancy (ISSHP), preeclampsia, transient gestational hypertension, and gestational hypertension (GH) are characterized by the new onset of hypertension (systolic blood pressure ≥140 mmHg or diastolic blood pressure ≥90 mmHg) at or after 20 weeks of gestation [4]. Normal blood pressure either in pre-pregnancy or in early pregnancy is important before its pregnancy-related decrease [4,5,6].

Gestational hypertension is hypertension arising de novo after 20 weeks of gestation in the absence of proteinuria and without biochemical or hematological abnormalities. It is usually not accompanied by fetal growth restriction. Outcomes in pregnancies complicated by gestational hypertension are normally good, but about a quarter of women with gestational hypertension (particularly those who present it before 34 weeks of gestation) will progress to preeclampsia and have poorer outcomes [4].

Preeclampsia is a multisystem disorder that manifests after the 20th week of gestation. It is characterized by hypertension and at least one of the following conditions: (1) proteinuria, (2) other maternal organ dysfunction, including acute kidney injury (AKI) (creatinine ≥90 μmol/L; 1 mg/dL), liver involvement (elevated transaminases, e.g., aspartate aminotransferase (AST) or alanine aminotransferase (ALT) > 40IU/L) with or without right upper quadrant or epigastric abdominal pain), neurological complications (examples include eclampsia, altered mental status, blindness, stroke, clonus, severe headaches, persistent visual scotomata), hematological complications (thrombocytopenia—platelet count below 150,000/μL, disseminated intravascular coagulation (DIC), (3) uteroplacental dysfunction (such as fetal growth restriction (FGR), abnormal umbilical artery Doppler wave form, or stillbirth) [4].

Proteinuria is not mandatory for a diagnosis of preeclampsia but is present in about 75% of cases. Rather, PE is diagnosed by the presence of de novo hypertension after 20 weeks of gestation accompanied by proteinuria and/or evidence of maternal acute kidney injury, liver dysfunction, neurological features, hemolysis or thrombocytopenia, and/or fetal growth restriction. Preeclampsia may develop or be recognized for the first time intrapartum or early post-partum in some cases [4].

Moreover, PE is characterized by reduced renal perfusion and damage to glomerular basement membrane resulting in leakage of proteins in the urine. Irrespective of the cause of hypertension, the quantification of proteinuria in pregnancy is vital not only for making a diagnosis but also for predicting maternal and fetal outcomes. Normally, pregnant women excrete minimal amounts of proteins in the urine (up to 150 mg/day) because of renal changes that occur during pregnancy; however, proteinuria >300 mg/day is considered abnormal for pregnant women.

Several methods for proteinuria evaluation are available in daily practice: (1) dipstick, (2) 24 h urine protein test, and (3) urine protein-to-creatinine ratio (UPCR). In the first method, urine test stripes are used. A standard test strip comprises up to 10 chemical pads that serve for the analysis of different parameters (e.g., proteins, pH, erythrocytes, leukocytes, nitrites, glucose, ketones). The test can be read within 60–120 s after dipping the stripe in a urine sample. The chemical pads change color after being immersed in the sample, and the results are interpreted by comparison of the pad color with the colors presented in the dipstick analysis guide. A 24 h urine protein test requires a 24 h collection of a urine sample. Then, one aliquot is taken from the total volume of the 24 h urine sample and tested. The UPCR, like the dipstick test is based on the analysis of a random urine sample, but like the 24 h urine protein test, it provides quantitative results. The UPCR has been validated for use as an estimator of 24 h urine protein levels from one random urine sample by analysis of the protein-to-creatinine ratio.

In clinical practice, proteinuria is assessed initially by automated dipstick urinalysis when possible; if this is not available, a careful visual dipstick urinalysis will suffice. If positive (≥‘1+’, 30 mg/dL), the UPCR should be determined. A UPCR ≥ 30 mg/mmol (0.3 mg/mg) is abnormal. A negative dipstick test can usually be accepted, and further UPCR testing is not required at that time. Proteinuria is not required for a diagnosis of pre-eclampsia. Massive proteinuria (>5 g/24 h) is associated with more severe neonatal outcomes [4].

Generally, the estimation of 24 h urine proteins has been considered the gold standard for quantitation of proteinuria [7]. However, the collection of a 24 h urine sample is inconvenient for the patient and prone to pre-analytical errors, thereby leading to inaccurate results or variations in the application of the assay [8]. Thus, the urine dipstick test or the UPCR is widely used to confirm proteinuria [6]. Recently, an appropriate cutoff value for the UPCR was set; however, the pros and cons of proteinuria testing are still debated [9,10]. Apparently, each analytical approach has some limitations, such as inaccuracy due to daily variations of protein secretion, falsely positive/negative results, and ease of sample testing [11,12,13].

In this study, we aimed to extensively evaluate proteinuria in women suspected of having GH or PE by comparing three available methods for protein testing: urine dipstick test, UPCR, and total protein in 24 h urine samples. We tested the analytical sensitivity and specificity and determined the cutoff values for UPCR in single mid-stream urine samples and total protein in 24 h urine samples. Moreover, we assessed biochemical parameters, such as serum creatinine, transaminases, and platelets, and evaluated their usefulness as additional markers of PE.

## 2. Materials and Methods

### 2.1. Study Design

This study included all patients with GH or PE (88 pregnant women) who were hospitalized between April 2015 and July 2017 at the Department of Obstetrics, Medical University of Gdansk, Poland. The patients were between 27 and 42 weeks of gestation with a singleton pregnancy with symptoms of GH or PE but without co-morbidities. Women with chronic secondary/essential hypertension, immunological diseases like Hashimoto’s disease, diabetes mellitus, pre-existing renal disease, intrauterine fetal death, multiple gestations, gestational diabetes and bacteriuria, multiple pregnancy, assisted reproductive technology in pregnancy, and premature rupture of membranes were excluded from the study. This study was approved by the Bioethics Committee at the Medical University of Gdansk (no. NKBBN/454/2014) and was conducted according to the principles of the Declaration of Helsinki. All participants provided a written informed consent to participate in the study.

### 2.2. Patients

Based on clinical and laboratory evaluations, according to the ISSHP classification, the patients were divided into two groups: GH group (*n* = 44) and PE group (*n* = 44). The study flow diagram is shown in Figure 1, and the baseline characteristics of the study population are provided in Table 1.

GH was defined as systolic blood pressure ≥140 mm Hg and diastolic blood pressure ≥90 mm Hg in a previously normotensive pregnant woman after 20 weeks of gestation without proteinuria or a sign of end-organ dysfunction. PE was diagnosed in patients with a high blood pressure (24 h respiratory rate records) and new-onset proteinuria, i.e., when resting blood pressure was ≥140/90 mmHg on two occasions that were at least 4 h apart, and significant proteinuria was detected in urine samples. In the absence of proteinuria, PE was diagnosed in women with hypertension in association with thrombocytopenia (platelet count <150,000/μL), impaired liver function (increased blood levels of liver aminotransferases to twice the normal concentration), new development of renal insufficiency (elevated serum creatinine >1.02 mg/dL), pulmonary edema, new-onset cerebral or visual disturbances, or uteroplacental dysfunction, including FGR. FGR was diagnosed as fetal abdominal circumference/estimated fetal weight <10th percentile combined with pulsatility index in the umbilical artery >95th percentile, or pulsatility index in the uterine artery >95th percentile, or abdominal circumference/estimated fetal weight <3rd percentile, or absent end-diastolic flow in the umbilical artery [14].

### 2.3. Methods

Proteinuria was assessed with the urine dipstick test, UPCR, and total protein in 24 h urine samples in each patient during hospitalization. The urine dipstick test was performed twice, and a positive result was considered significant. Subsequently, creatinine, AST, ALT, and complete blood count (CBC) were evaluated. Serum and urine biochemical parameters were assayed with Architect analytical system (Abbott) and CBC with Sysmex analytical system (Sysmex). The urine dipstick test was performed with the Iris urinalysis system (Beckman Coulter).

### 2.4. Statistical Analysis

All analyses were performed using Statistica 11.0 (Statsoft). Between-group differences were determined using nonparametric tests. Significance was set at *p* < 0.05. A receiver operating characteristic (ROC) analysis was performed to verify the diagnostic usefulness of the evaluated markers, and data are presented with 95% confidence interval (CI). Data are expressed as medians with interquartile range. Heat map and principal component analyses were performed in ClustVis, based on the Euclidean distance between clusters [15].

## 3. Results

### 3.1. Proteinuria Testing

Three different methods were used for proteinuria evaluation: urine dipstick test, UPCR, and total protein in a 24 h urine sample. In 9% (4/44) of the patients in the PE group, the urine dipstick test was falsely negative in the first test, while the second test revealed a positive result. In the GH group, 25% (11/44) of the patients had falsely positive results in the urine dipstick test; UPCR and 24 h urine test did not confirm the dipstick test results, but all three tests were performed. Moreover, UPCR (median 10683.00 vs. 16.41 [mg/mmol]; *p* = 5.21 × 10^−10^; Mann–Whitney U test) and total protein in a 24 h urine specimen (median 0.98 vs. 0.16 [g/24 h]; *p* = 4.38 × 10^−6^; Mann–Whitney U test) were higher in the PE group. Each positive UPCR result corresponded to a significant total protein value in the 24 h urine specimen. The UPCR cutoff value of 30 mg/mmol was validated as significant for PE, based on the ROC analysis. While the area under curve was 0.96 (*p* <0.0001; 95% CI, 0.921–1.004), the sensitivity and specificity were 89% (95% CI, 75–97) and 100% (95% CI, 87–100), respectively. In the 24 h urine protein test, the area under the curve, sensitivity, and specificity were 0.94 (*p* <0.0001; 95% CI, 0.881–1.008), 80% (95% CI, 61–92), and 100% (95% CI, 88–100), respectively, for the cutoff value of 0.26 g/24 h (Figure 2).

### 3.2. General Biochemistry Testing

The CBC results showed that patients with PE and GH did not differ in whole-blood platelet (PLT) count (median 213.00 vs. 220.00 [10^9^/L]; *p* = 0.612; Mann–Whitney U test) and hematocrit (HCT) (median 44.90 vs. 35.20 [%]; *p* = 0.08; Mann–Whitney U test); however, the GH group had a higher hemoglobin (Hgb) concentration (median 12.60 vs. 11.90 [g/dL]; *p* = 0.012; Mann–Whitney U test). Moreover, serum AST (median 21.50 vs. 16.00 [U/L]; *p* = 4.87 × 10^−5^) and ALT (median 16.50 vs. 11.00 [U/L]; *p* =1.42 × 10^−3^; Mann–Whitney U test) concentrations increased in the PE group compared to those in the GH group, but not serum creatinine (median 0.69 vs. 0.68 [mg/dL]; *p* = 0.269; Mann–Whitney U test).

ROC analysis was performed to determine the parameters’ sensitivity and specificity and to validate them as markers of PE. The area under curve, sensitivity, and specificity were 0.536 (*p* = 0.559; 95% CI, 0.413–0.659), 57% (95% CI, 41–72), and 45% (95% CI, 30–61), respectively, for PLT; 0.630 (*p* = 0.035; 95% CI, 0.512–0.748), 61% (95% CI, 46–76), and 66% (95% CI, 48–78), respectively, for HCT; and 0.657 (*p* = 0.011; 95% CI, 0.542–0.772), 61% (95% CI, 46–76), and 59% (95% CI, 43–74), respectively, for Hgb. The area under curve, sensitivity, and specificity were 0.753 (*p* = 0.0001; 95% CI, 0.646–0.860), 70% (95% CI, 53–84), and 66% (95% CI, 49–80), respectively, for ALT; 0.700 (*p* = 0.002; 95% CI, 0.585–0.814), 73% (95% CI, 56–86), and 61% (95% CI, 45–76), respectively, for AST; and 0.569 (*p* = 0.270; 95% CI, 0.447–0.692), 60% (95% CI, 43–75), and 48% (95% CI, 32–63), respectively, for creatinine (Figure 3).

Neither principal component analysis nor heat maps with clustering revealed differences in the collectively analyzed data between patients with PE and GH (Figure 4).

## 4. Discussion

In this study, we assessed the utilization of proteinuria and biochemical parameters as markers of PE.

The urine dipstick test is a semi-quantitative assay, while the UPCR and total protein in the 24 h urine test allow quantitative determinations. The urine dipstick test is a screening assay, which could detect positive cases (true disease). However, our data showed that 9% of patients were misdiagnosed as not having proteinuria using a single urine dipstick test, and a second sample testing was required. By contrast, the UPCR and total protein in a 24 h urine sample could fully distinguish patients with PE and GH, and a second test was not needed. Moreover, urine dipstick test results may vary depending on the maternal hydration status. Thus, even trace proteinuria may be reported as significant if the mother is dehydrated, and in contrast, proteinuria may be missed if the mother is overhydrated. In addition, the test readout could be altered depending on the test features (limit of detection and limit of quantification), alkalinity of the urine, and presence of infections. This is important, as proteinuria must be confirmed or excluded in pregnant women presenting with hypertension in order to diagnose PE.

In our cohort, both UPCR and total protein in the 24 h urine test were validated as adequate for PE diagnosis. Both their sensitivity and their specificity were good enough to appropriately identify patients with PE or GH. However, while the 24 h urine test is considered the gold standard for proteinuria testing, the test itself is cumbersome, time-consuming, and prone to pre-analytical errors, which could in turn results in lack of compliance by and inconvenience to patients. Also, resting in the supine position during hospitalization may result in urine stagnation in the renal system, and the amount of collected urine may not reflect the actual 24 h volume. Hence, UPCR is a potentially useful marker of PE. As previously shown, UPCR was not affected by variations in urine concentration and the amount of urine excreted in 24 h [16]. Our study demonstrates that UPCR is as valuable as the 24 h urine test in patients with preeclampsia, and a UPCR cutoff of 30 mg/mmol coincides with proteinuria in the corresponding 24 h urine sample.

Creatinine was not adequate for PE diagnosis; its levels were comparable in patients with PE and with GH. Such a finding is apparently surprising, as one could suspect that protein loss due to altered renal filtration coexists with increased serum creatinine. Nevertheless, stable creatinine levels reflect adaptive renal capacities, and increased creatinine is more specific to kidney injury, while irreversible damage causes glomerulonephritis.

Although AST, ALT, and PLT levels increased in the PE compared to the GH group, none of these values is useful for PE diagnosis based on the ROC analysis. Similar to creatinine, this finding may be because of early diagnoses in our cohort and lack of complications due to prolonged hypertension and proteinuria or signs of end-organ dysfunction. Collectively, the early diagnosis of protein loss due to hypertension prevented the subsequent development of full-blown preeclampsia with clinical symptoms and increased markers of tissue damage.

The significant correlation between UPCR and proteinuria in 24 h urine samples is a noteworthy finding of our study. While the latter represents the gold standard for the diagnosis of proteinuria, it delays diagnosis by 24 h, is not well tolerated by the patients, and cannot be performed in an emergency room setting. When identifying markers of PE, not only sensitivity and specificity should be considered but also turnaround time from sampling to results, which may in turn lead to timelier decision-making, likely reduce patients’ anxiety, shorten the length of hospital stay, thereby minimizing the associated cost, and help “target” women with a real pathology for treatment.

In practice, the 24 h urine protein measurement could mostly be replaced by a spot test determining urine protein/creatinine ratio, with a value ≥30 mg per mmol (= 0.26 mg/mg, usually ’rounded’ to 0.3 mg/mg) representing significant proteinuria. This would eliminate the inherent difficulties in undertaking 24 h urine collections and speed up the process of decision-making [4]. At present, there are insufficient data to recommend using the urinary albumin/creatinine ratio, but this may change when more research becomes available, such as the results of the DAPPA (Diagnostic Accuracy in Preeclampsia using Proteinuria Assessment, RCTN82607486) clinical trial [4,17,18,19].

The determination of the protein/creatinine ratio would be a valuable tool for the early diagnosis of preeclampsia if its accuracy is acceptable. It could prevent unnecessary hospitalizations and testing and allow earlier diagnoses [20]. Several studies have evaluated the usefulness of the protein-to-creatinine ratio as a screening tool for the determination of proteinuria in subjects with suspected preeclampsia [17]. Sanchez’s study, as well as three systematic reviews with meta-analyses, have shown that the protein-to-creatinine ratio correlates well with the results of a subsequent 24 hour urine test and is useful for the diagnosis of significant proteinuria [17,18,20,21].

Although the 24 h urine collection is still accepted as the gold standard for the diagnosis of proteinuria during pregnancy, some authorities have suggested that the protein-to-creatinine ratio should be employed as the preferred method of quantifying proteinuria. It is recommended to use the 24 h urine test for subjects with an abnormal protein-to-creatinine ratio [17]. In addition, Waugh et al. concluded that the collection of a 24 h urine sample has comparable value to that of a spot urine sample to quantify proteinuria in women with hypertension in pregnancy [22]. Therefore, these results from Waugh’s study do not support the recommendation of a 24 h urine sample collection for hypertensive pregnant women. Once the UPCR is confirmed to be >30 mg/mmol, no further proteinuria measurements are required during hypertensive pregnancies. A randomly determined urine protein-to-creatinine ratio provides useful evidence to rule out the presence of significant proteinuria in patients at risk for preeclampsia [16,22].

## 5. Conclusions

Although the 24 h urine test for total protein remains the gold standard for proteinuria evaluation, it has several limitations. The dipstick test has insufficient specificity and sensitivity; nevertheless, because of its simplicity and cost, it may still be used as a primary screening method but is not suitable for the quantitation of proteinuria. In comparison, the UPCR is a reliable, relatively faster, and equally accurate method for the detection and quantitation of proteinuria; it correlates well with the 24 h urine protein estimation and, thus, could be used as an alternative to 24 h urine testing for patients suspected of preeclampsia. Furthermore, the UPCR test can be performed in emergency room settings and is more clinically useful for the assessment of proteinuria in pregnant women with hypertension.

## Figures and Tables

**Figure 1 ijerph-17-04195-f001:**
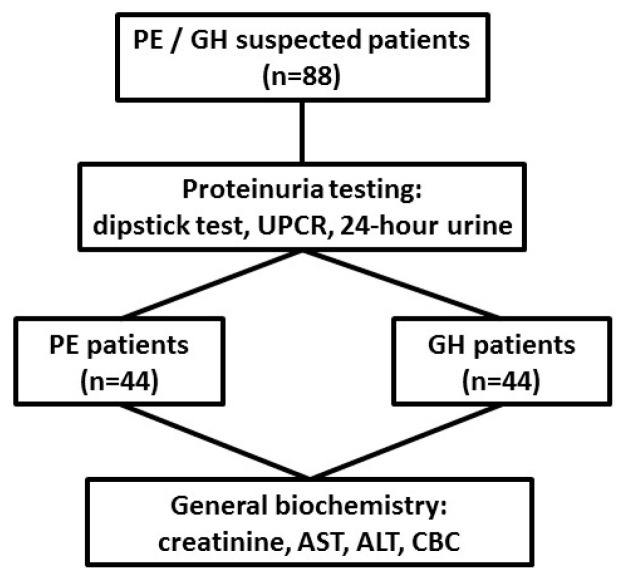
Study flow diagram. Patients were classified into two groups based on clinical features and proteinuria assessment: preeclampsia (PE) group (*n* = 44) or gestational hypertension (GH) group (*n* = 44), and general biochemistry tests were performed. UPCR: urine protein-to-creatinine ratio, AST- aspartate aminotransferase: ALT-alanine aminotransferase: CBC-complete blood count.

**Figure 2 ijerph-17-04195-f002:**
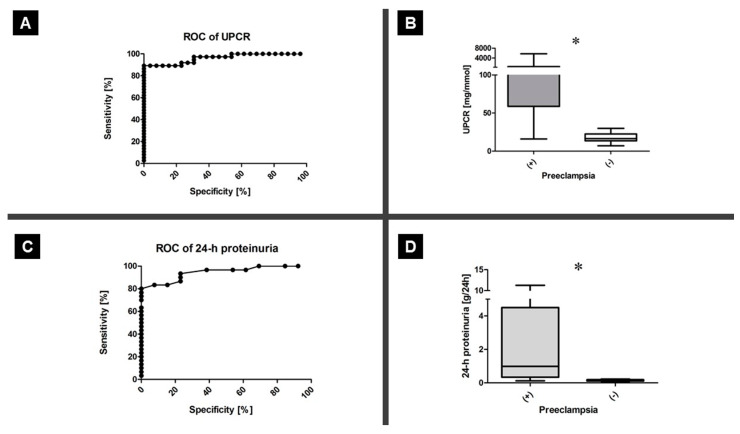
Proteinuria testing. Protein loss was assessed by determining the UPCR and performing the 24 h urine test. The UPCR cutoff of 30 mg/mmol was adequate for the diagnosis of preeclampsia (**A**). The UPCR values increased in the PE group (**B**). For the cutoff value of 0.26 g/24 h, the total protein level in 24 h urine samples was adequate for the diagnosis of PE (**C**). Total protein was increased in the PE group (**D**); * *p* < 0.05.

**Figure 3 ijerph-17-04195-f003:**
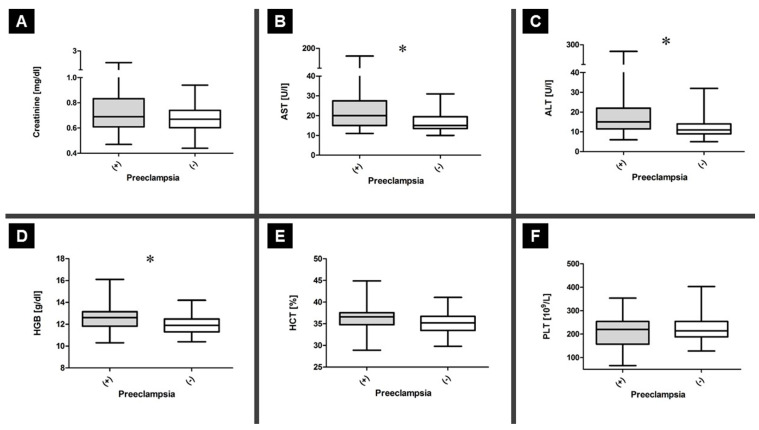
General biochemistry testing. The levels of creatinine (**A**), AST (**B**), ALT (**C**), hemoglobin (Hgb0) (**D**), hematocrit (HCT) (**E**), and platelet count (PLT) (**F**) were compared between the PE and GH groups. Significant differences in AST, ALT, and PLT (**F**) were compared between the PE and the GH groups. Significant differences in AST, ALT, and platelet count were noted and indicated with “*”; * *p* < 0.05.

**Figure 4 ijerph-17-04195-f004:**
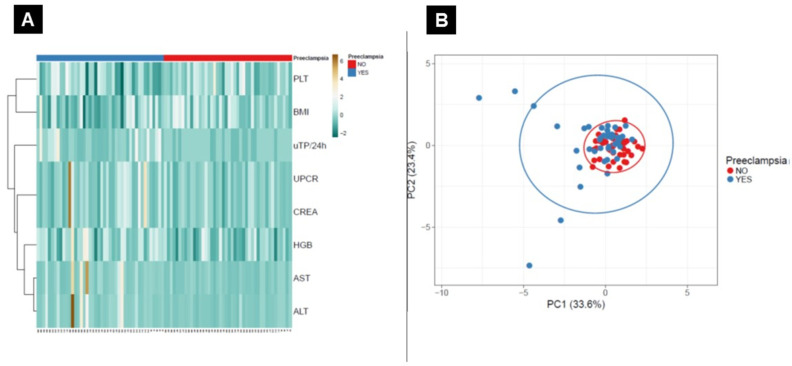
Heat maps and principal component analysis (PCA) analysis. Heat maps based on the Euclidean distance between clusters for patients with PE and GH (**A**). PCA graph for the PE and GH groups (**B**). The *X* and *Y* axes show principal component 1 and principal component 2, which explain 33.6% and 23.4% of the total variance, respectively.

**Table 1 ijerph-17-04195-t001:** Patient characteristics.

Patients Status	PE *n* = 44	GH *n* = 44	*p ^a^*
Age (years, mean ± SD)	28 ± 4.05	30 ± 4.60	0.124
Period of gestation	35	39	2.469 × 10^−6^
Body mass index (kg/m^2^) (median, min/max)	30 (21/46)	33 (26/42)	7.013 × 10^−8^
Parity			
0	36	34	NT
1	7	7	NT
>1	1	3	NT

*^a^* χ^2^ test of association or Mann–Whitney U-test comparing PE vs. GH; NT—not tested.

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
