# Peer review of "Comparisons of Dipstick Test, Urine Protein-to-Creatine Ratio, and Total Protein Measurement for the Diagnosis of Preeclampsia"

_ijerph, 2020, doi:10.3390/ijerph17124195_

Round 1

Reviewer 1 Report

This study compared the three detection approaches, dipstick test, urine protein-to-creatine ratio (UPCR) and total protein measurement in 24h urine samples for diagnosis of preeclampsia. About 88 patients were included in the study. It is suggested that UPCR is a reliable and accurate approach for quantification of proteinuria for preeclampsia diagnosis, which could be potentially used as an alternative to 24h total protein measurement. The following comments should be addressed before the paper can be further considered:

  • The title of manuscript is too long and confusing, which should be shortened. It can be changed to “Comparisons of dipstick test, urine protein-to-creatine ratio and total protein measurement for diagnosis of preeclampsia”
  • In introduction, page 2, the sentence “preeclampsia is gestational hypertension….” is incorrect. Preeclampsia is not a hypertension but can be characterized or diagnosed with hypertension. Please revise the sentence.
  • In introduction, three existing detection approaches (i.e., dipstick test, UPCR and total protein measurement) should be briefly introduced. How are they performed? Does UPCR involve immediate urine sample collection? The pros and cons of these three approaches should also be discussed.
  • Figure 1, does “GE patients” mean “GH patients”?
  • Figure 2 & Figure 3, the p value should be defined. Is p<0.05 statistically significant?
  • In discussion, page 7, since preeclampsia has been introduced in the introduction, it does not need to be introduced again in the discussion. The discussion section should be more concise.
  • In discussion, page 8, the term “in conclusion” should be removed. The summary should only be written in the conclusion section.

Author Response

  • Point 1: The title of manuscript is too long and confusing, which should be shortened. It can be changed to “Comparisons of dipstick test, urine protein-to-creatine ratio and total protein measurement for diagnosis of preeclampsia”
  • We would like to thank the Reviewer for this suggestion. We changed the title accordingly to: „Comparisons of dipstick test, urine protein-to-creatine ratio and total protein measurement for diagnosis of preeclampsia” line 2-4
  • Point 2: In introduction, page 2, the sentence “preeclampsia is gestational hypertension….” is incorrect. Preeclampsia is not a hypertension but can be characterized or diagnosed with hypertension. Please revise the sentence.

We are sorry for this inaccurate phrase. We agree with the Reviewer and corrected the sentence. Please see corrected version of the “manuscript- with changes highlighted” page 2, line 54.

Point 3: In introduction, three existing detection approaches (i.e., dipstick test, UPCR and total protein measurement) should be briefly introduced. How are they performed? Does UPCR involve immediate urine sample collection? The pros and cons of these three approaches should also be discussed.

  • We would like to thank the Reviewer for this comment. We agree that the results of the study will be more clear for the readers after consistent introduction of these methods. Please see corrected version of the “manuscript- with changes highlighted” page 2-3, line 74-84
  • Point 4: Figure 1, does “GE patients” mean “GH patients”?

We would like to thank the Reviewer for this comment. We have already corrected this typo. Please see Figure 1.

  • Point 5: Figure 2 & Figure 3, the p value should be defined. Is p<0.05 statistically significant?
  • Yes, we confirm that p<0.05 was considered statistically significant. We have added this phrase in the figure legends to avoid confusion.
  • Point 6: In discussion, page 7, since preeclampsia has been introduced in the introduction, it does not need to be introduced again in the discussion. The discussion section should be more concise.
  • We would like to thank the Reviewer for this comment. We have corrected discussion accordingly to make the text more readable. Please see the discussion section in the corrected version of the “manuscript- with changes highlighted” page 7-8, line 213, 256
  • Point 7: In discussion, page 8, the term “in conclusion” should be removed. The summary should only be written in the conclusion section.

We rephrased the text according to the Reviewer suggestion. Please see corrected version of the “manuscript- with changes highlighted” page 7-8, line 256; 282

Reviewer 2 Report

The manuscript titled ‘Not urine dipstick test and serum creatinine but urine protein to creatinine ratio and protein in 24-hour urine sample are superior for diagnosis of preeclampsia’ compares the analytical sensitivity and specificity of three different tests including urine dipstick, urine protein to creatinine ratio (UPCR) and protein in 24h urine in the diagnosis of preeclampsia in pregnant women and analyzes the pros and cons of these tests, demonstrating urine protein to creatinine ratio is a more reliable method for quantification of proteinuria compared to urine dipstick, and well related with 24h urine protein estimation, which can be used as an alternative to the gold standard test. Meanwhile authors also test general biochemistry parameters including serum creatinine, AST, ALT, hematocrit, etc. and illustrate that these parameters are not directly correlated to preeclampsia, but only alert early protein loss due to hypertension, not yet development of preeclampsia. I think this is more like a supplementary research as there have been a lot of studies on this topic and also this method has been well established. It has also determinesd the criteria of urine protein-to-creatinine ratio for the diagnosis of preeclampsia.

Comments:

  1. As mentioned above, what do authors think is the originality of this research since there are lots of preclinical and clinical studies regarding this topic. (Authors also cited these studies in Line 269-275 and stated ‘Several studies have evaluated the usefulness of the protein-to-creatinine ratio as a screening tool for the evaluation of proteinuria in subjects with suspected preeclampsia’ and ‘Sanchez’s study, as well as 3 systematic reviews with meta-analysis, have shown that the protein-to- creatinine ratio correlates well with subsequent 24-hour urine collection and is useful for the diagnosis of significant proteinuria’)

  1. Authors mentioned in 3.1. Proteinuria testing – three different methods were used proteinuria evaluation. When using urine dipstick test, 4 in 44 in the PE group were tested falsely negative and the second test revealed positive while 11 in 44 in the GH group had accidently positive results so these 11 patients did not follow the UPCR and total protein in a 24h urine test.

My question is, what indeed determines these suspected patients as PE or GH? Total protein in a 24h urine test? How authors state that 4 in the PE group are falsely negative and 11 in the GH group are accidently positive since these 11 patients did not follow the UPCR and total protein in a 24h urine tests? In this case there will be only 33 patients in the GH group rather than 44 in Figure 1 for UPCR and total protein in a 24h urine tests? By the way, in Figure 1, GH, not GE.

  1. Line 174-176, serum ALT (median 21.5 vs 11) and AST (median 16.5 vs 16) concentrations decrease in the PE group compared to those in the GH groups? Line 239-240, authors claimed ‘Although AST, ALT, and PLT count increased in the PE compared to the GH group.’ According to Figure 3, AST and ALT increase in the PE group. On the other hand, the median is not consistent with that in Figure 3. ALT in two groups should be close rather than 21.5 vs 11, while AST median in PE group is close to 20? Is there anything misunderstanding?

  1. Line 214-215, ‘9% of patients were misdiagnosed as having PE’? As I understand, these 9% patients are PE positive, correct? Confusing. Do authors mean 9% of patients having PE were misdiagnosed?

  1. Line 279-281, ‘The collection of a 24-hour urine sample confers no additional value over a spot urine sample to quantify proteinuria in women with hypertension in pregnancy’. My question may be not professional – is the UPCR test time-specific or random/spot? If it is time-specific, what time point would it be to deliver correct result for diagnosis? Otherwise, the UPC ratio is stable all day whenever testing?

  1. What is the relationship between UPCR and total protein in 24h urine test since they are well correlated? Is it convertible or predictable from one to another?

Minor Comments:

  1. Line 40-43, repeat sentence. Could authors simplify? ‘PE typically affects 2–5% of pregnant women and is one of the leading causes of maternal and perinatal morbidity and mortality, especially when the condition is of early onset [1,2]. Moreover, PE is an important cause of maternal morbidity and mortality and a significant contributor to higher incidences of perinatal morbidity and mortality.’
  2. Line 79, ‘performer’?
  3. Line 198, Figure 4
  4. Line 204, ‘include’ and ‘requires’?
  5. Line 222, ‘what announce PE’?
  6. Line 240, ‘was’?
  7. Line 242, ‘state’?

Above all, besides addressing these comments, I suggest that authors put more original research content or have more detailed experiments, analysis or discussions to qualify for publication.

Author Response

Reviewer 2:

  1. As mentioned above, what do the authors think is the originality of this research since there are lots of (many) preclinical and clinical studies regarding this topic. (the authors also cited these studies in Line 269-275 and stated ‘Several studies have evaluated the usefulness of the protein-to-creatinine ratio as a screening tool for the evaluation of proteinuria in subjects with suspected preeclampsia’ and ‘Sanchez’s study, as well as 3 systematic reviews with meta-analysis, have shown that the protein-to- creatinine ratio correlates well with subsequent 24-hour urine collection and is useful for the diagnosis of significant proteinuria’)

 We would like to thank the Reviewer for these remarks. We agree with the Reviewer that UPCR is not a novel issue in PE research, but this paper will have an impact on the recommendations in a routine clinical practice and it will be useful for designing diagnostic algorithm. In many countries dipstick is used as a first choice test for rapid detection of patients with proteinuria. In majority of cases negative result is not verified with other methods. No recommendations exist if dipstick or UPCR should be performed for the first line screening when proteinuria is suspected. There are neither clear guidelines which method should be chosen nor a laboratory algorithm on how to confirm proteinuria. For this reason, our paper expands the knowledge regarding proteinuria testing and gives evidence that proteinuria cannot be confirmed based on a simple dipstick test alone and that further diagnostics are needed. With this study we also show that replacement of dipstick tests with UPCR for fast screening for proteinuria will give much more reliable results which is of great importance when clinical decisions are made. Therefore, our study may have a serious impact on daily clinical practice, which is why we think that our paper is pertinent and interesting for a wide physician readership. At this point we also would like to add that screening test should detect all the positive results even at the expense of false positive results. As in contrary to UPCR, dipstick can give false negative results, it should be avoided in daily clinical practice.

The authors mentioned in 3.1. Proteinuria testing – three different methods were used in proteinuria evaluation. When using a urine dipstick test, 4 of  44 in the PE group (were tested falsely negative and the second test revealed positive) unclear wording while 11 of  44 in the GH group had accidently positive results so these 11 patients did not follow the UPCR and total protein in a 24h urine test.My question is, what indeed (precisely) determines these suspected patients as PE or GH? Total protein in a 24h urine test? How (can the) authors state that 4 in the PE group are falsely negative and 11 in the GH group are accidently positive since these 11 patients did not follow the UPCR and total protein in a 24h urine tests? In this case there will (would) be only 33 patients in the GH group rather than 44 in Figure 1 for UPCR and total protein in a 24h urine tests, By the way, in Figure 1, GH, not GE.

We would like to thank the Reviewer for these specific remarks. There was an error in the manuscript. In case of 11 out of 44 patients in the GH group it should be stated that UPCR and 24h urine test did not confirm the dipstick test results, but all 3 tests were performed. Therefore, GH group indeed comprised of 44 patients. We have also corrected the Figure 1 according to the Reviewer comment (GE was replaced with GH), please see corrected version of the “manuscript- with changes highlighted” page 3. Responding to the Reviewer doubts PE and GH were determined according to total protein levels in a 24h urine test. This method was treated as a gold standard.

This was the major aim of our study.

  1. Line 174-176, serum ALT (median 21.5 vs 11) and AST (median 16.5 vs 16) concentrations decrease in the PE group compared to those in the GH groups? Line 239-240, authors claimed ‘Although AST, ALT, and PLT count increased in the PE compared to the GH group.’ According to Figure 3, AST and ALT increase in the PE group. On the other hand, the median is not consistent with that in Figure 3. ALT in two groups should be close rather than 21.5 vs 11, while AST median in PE group is close to 20? Is there anything misunderstanding?

  We would like to thank the Reviewer for the comment and we are sorry for this mistake. We confirm that the median value of AST and ALT was higher in PE compared with GH. The error has been corrected. Please see corrected version of the “manuscript- with changes highlighted” page 6, line 184-185:

  1. Line 214-215, ‘9% of patients were misdiagnosed as having PE’? As I understand, these 9% (of) patients are PE positive, correct? Confusing. Do authors mean 9% of patients having PE were misdiagnosed?

We would like to thank the Reviewer for this comment. We have to admit that it was our error again. The 9% of patients were misdiagnosed as not having proteinuria, but they were PE positive anyway. PE was diagnosed according to other factors, but not proteinuria in these patients.  Please see corrected version of the “manuscript- with changes highlighted” page 7, line 217-218.

Line 279-281, ‘The collection of a 24-hour urine sample confers no additional value over a spot urine sample to quantify proteinuria in women with hypertension in pregnancy’. My question may be not professional – is the UPCR test time-specific or random/spot? If it is time-specific, what time point would it be to deliver correct result for diagnosis? Otherwise, the UPC ratio is stable all day whenever testing?

  We would like to thank the Reviewer for this comment. In our opinion the question is fully justified. However, in this particular sentence we have directly cited the paper of Waugh J et al published in 2017 in J. Health Technol Assess. Therefore, this is not our point of view, but just a subjective opinion of the other group. Generally, UPCR is considered a stable parameter. Stable enough to be used for PE diagnosis. However, some factors may affect both creatinine and protein testing in urine. .Intra-individual variability in creatinine excretion may be due to differences in meat ingested or physical activity. The last one may also affect urinary protein concentration in healthy individuals. , Therefore, 24h urine test is still a gold standard, but in cases where fast information regarding proteinuria is required UPCR, instead of dipstick should be chosen. We have corrected the statements in the discussion section to make them less confusing. Please see corrected version of the “manuscript- with changes highlighted” page 8, line 273-275.

  1. What is the relationship between UPCR and total protein in 24h urine test since they are well correlated? Is it convertible or predictable from one to another?

Both UPCR and 24h urine tests reflect protein excretion. They are not convertible, but predictable. The two tests differ in the way how amount of protein content in urine is calculated. From sample to sample the volume and density of urine differ. This can be standardized when the protein concentration is calculated in relation to creatinine (in UPCR) or when urine is collected during a fixed and long enough time (24 h).

Minor Comments:

  1. Line 40-43, repeat sentence. Could authors simplify? ‘PE typically affects 2–5% of pregnant women and is one of the leading causes of maternal and perinatal morbidity and mortality, especially when the condition is of early onset [1,2]. Moreover, PE is an important cause of maternal morbidity and mortality and a significant contributor to higher incidences of perinatal morbidity and mortality.’
  2. Line 198, Figure 4
  3. Line 204, ‘include’ and ‘requires’?
  4. Line 222, ‘what announce PE’?
  5. Line 240, ‘was’?
  6. Line 242, ‘state’?

 We would like to thank the Reviewer for these comments. We have corrected all these errors in the text.   Please see corrected version of the “manuscript- with changes highlighted”

  1. Page 1 line 37-41 2 -Line 208; 3- removed part of the text ; 4- line 224-225. 5-243

6- line 245.

Above all, besides addressing these comments, I suggest that authors put more original research content or have more detailed experiments, analysis or discussions to qualify for publication.

We are grateful for all the comments and remarks of the Reviewer. We have corrected the text accordingly. We believe that after implementation of new information, paragraphs and correction of some errors we managed to improve the paper quality and made it clear for the readership. We also believe that this paper reveals observations that are important for the clinical practice of PE diagnostics. This is especially critical for those institutions which lack strict laboratory criteria how to test proteinuria in PE suspected women. We hope that in the current version the manuscript is acceptable for the publication in the International Journal of Environmental Research and Public Health.

Round 2

Reviewer 1 Report

The authors have addressed my comments. I have no further comments.

Reviewer 2 Report

Thanks for the detailed explanations and careful revisions. Obviously, the new title is more accurate and suitable to summarize the research and make this study outstanding from the previous ones. As a guideline for daily clinical practice, I think the manuscript is valuable for physicians and clinical community related to obstetrics and gynecology.

Minor Comments:

  1. in Abstract and Figure 4 legend, replace ‘GE’ with ‘GH’.

I recommend the revised manuscript for publication in International Journal of Environmental Research and Public Health.